# Ultrasound Image Classification of Thyroid Nodules Using Machine Learning Techniques

**DOI:** 10.3390/medicina57060527

**Published:** 2021-05-24

**Authors:** Vijay Vyas Vadhiraj, Andrew Simpkin, James O’Connell, Naykky Singh Ospina, Spyridoula Maraka, Derek T. O’Keeffe

**Affiliations:** 1School of Medicine, College of Medicine Nursing and Health Sciences, National University of Ireland Galway, H91 TK33 Galway, Ireland; jamesoconnell118@gmail.com (J.O.); derek.okeeffe@nuigalway.ie (D.T.O.); 2Health Innovation Via Engineering Laboratory, Cúram SFI Research Centre for Medical Devices, Lambe Institute for Translational Research, National University of Ireland Galway, H91 TK33 Galway, Ireland; 3School of Mathematics, Statistics and Applied Maths, National University of Ireland, H91 TK33 Galway, Ireland; andrew.simpkin@nuigalway.ie; 4Division of Endocrinology, Department of Medicine, University of Florida, Gainesville, FL 3210, USA; naykky.singhospina@medicine.ufl.edu; 5Division of Endocrinology and Metabolism, University of Arkansas for Medical Sciences, Little Rock, AR 72205, USA; smaraka@uams.edu; 6Medicine Section, Central Arkansas Veterans Healthcare System, Little Rock, AR 72205, USA; 7Lero, SFI Centre for Software Research, National University of Ireland Galway, H91 TK33 Galway, Ireland

**Keywords:** computer aided diagnostics, CAD, artificial intelligence, AI, digital health, TI-RADS, big data, ANN, SVM, malignant, benign, cancer

## Abstract

*Background and Objectives*: Thyroid nodules are lumps of solid or liquid-filled tumors that form inside the thyroid gland, which can be malignant or benign. Our aim was to test whether the described features of the Thyroid Imaging Reporting and Data System (TI-RADS) could improve radiologists’ decision making when integrated into a computer system. In this study, we developed a computer-aided diagnosis system integrated into multiple-instance learning (MIL) that would focus on benign–malignant classification. Data were available from the Universidad Nacional de Colombia. *Materials and Methods*: There were 99 cases (33 Benign and 66 malignant). In this study, the median filter and image binarization were used for image pre-processing and segmentation. The grey level co-occurrence matrix (GLCM) was used to extract seven ultrasound image features. These data were divided into 87% training and 13% validation sets. We compared the support vector machine (SVM) and artificial neural network (ANN) classification algorithms based on their accuracy score, sensitivity, and specificity. The outcome measure was whether the thyroid nodule was benign or malignant. We also developed a graphic user interface (GUI) to display the image features that would help radiologists with decision making. *Results*: ANN and SVM achieved an accuracy of 75% and 96% respectively. SVM outperformed all the other models on all performance metrics, achieving higher accuracy, sensitivity, and specificity score. *Conclusions*: Our study suggests promising results from MIL in thyroid cancer detection. Further testing with external data is required before our classification model can be employed in practice.

## 1. Introduction

Evaluating thyroid nodules is clinically challenging. Thyroid nodules are frequently detected incidentally during the diagnostic imaging of the neck [1]. They may be found in 42–76% of people, becoming more prevalent with increasing age [2]. Most thyroid nodules are benign, but 10% may be malignant [3]. Thyroidectomy, radioactive iodine therapy, immunotherapy, or chemoradiotherapy to prevent recurrence and death may be indicated for thyroid cancer treatment depending on the histological subtype, patient preference, and comorbidities [1,3,4].

A fine needle aspirate biopsy (FNAB) of the nodule may be obtained for a cytological examination of the nodule. As well as generating anxiety for patients [5], FNABs may cause localized pain and although generally safe, there is a low risk of hematoma [6]. In 20–30% of FNABs, the outcome of cytological examination is indeterminate, which means they do not always yield clinically useful results [7,8]. Additionally, some older patients with comorbidities may not suffer adverse outcomes within their lifetime from low-risk thyroid cancers detected after the FNAB of a thyroid nodule [9]. In South Korea, thyroid cancer screening led to a sharp increase in the incidence of papillary thyroid cancer without any impact on thyroid cancer mortality [10], raising concerns for the overdiagnosis of thyroid cancer. As an alternative to invasive FNAB, a conservative strategy such as the active ultrasound surveillance of thyroid nodules may be opted for in carefully selected patients, but this strategy risks missing clinically relevant cancers [11].

Evidently, when deciding the best approach to managing a thyroid nodule, clinicians and patients must balance the risks of an FNAB and overdiagnosis against those of misdiagnosis. To risk stratify nodules and aid in decision making with regard to the need for an FNAB, the American College of Radiology as shown in Appendix C, Table A1 and the Korean Society of Thyroid Radiology each have developed their own Thyroid Imaging Reporting and Data Systems (TI-RADS).

Based on nodule ultrasonographic characteristics such as internal composition, echogenicity, calcification, margins, and size, radiologists can report on the risk of thyroid nodules being malignant using a standardized scoring system [5]. The disadvantages of TI-RADS are that by grouping nodules with different risk factors into a small number of categories, they are managed as equivalent despite their risk of malignancy being potentially very different [5]. Inter-observer and intra-observer variability may also be challenging when implementing TI-RADS [12]. This could potentially cause variability in treatment decisions and outcomes in different patients despite having similar tumors [13].

Systems, which can reliably identify patients with benign and malignant thyroid nodules, while avoiding an invasive FNAB and its associated negative consequences for patients as well as avoiding resource utilization by healthcare providers, are needed. Computer aided diagnosis (CAD) may be of practical application here. Described TI-RADS features may be integrated into computer systems, which can be used to aid in the radiological classification of thyroid nodules. With a structured approach for image acquisition, feature extraction, classification, training, and prediction using machine learning (supervised and unsupervised), it is possible to build and train a model that would predict whether thyroid nodules are malignant or benign [13]. Furthermore, with the use of the same model, a graphic user interface (GUI) can display nodule image features to aid in the decision-making process of radiologists. The accuracy score, sensitivity, and specificity of the models would be important determinants of the clinical utility of these systems. The aim of this research was to build a CAD model that would predict whether thyroid nodules are benign or malignant.

### Previous Related Work

For this study, a comprehensive literature review, which included 35 research papers on the ultrasound of thyroid nodules using the CAD system, was completed. Numerous CAD systems have been studied for automated thyroid detection in recent years. Most CAD systems are based on the Convolution Neural Network (CNN) and the Radial basis function (RBF) neural network, which use K-NN for high accuracy scores. Moreover, the multitask cascade convolution neural network (MC-CNN) requires a more extensive data set for a training class [14,15,16,17,18,19,20].

Given that there was a lack of research on CAD systems based on the Artificial neural network (ANN) (Figure 1), this was chosen as the primary model for this research. Other neural networks featured in the literature were Probabilistic neural networks (PN) and Multilayer Perceptron Neural Network (MLPNN) [1,2,21,22,23,24,25,26,27].

The selection of classifiers mainly depends on how well the model is trained and tested. Figure 2 describes the frequency of the different classifiers used in the literature. SVM is widely used because of its flexible kernel function and threshold. A total of 28% of the research conducted on thyroid nodule detection is based on the SVM algorithm. However, only 2% of the research conducted was based on ANN and SVM. Therefore, ANN and SVM were chosen as the classifiers for this research.

Regarding image processing and segmentation, median filters are more practical than mean filters or binarization methods when converting greyscale images into black and white. Therefore, these methods were adopted. In computer vision, for any complicated computational task, feature extraction is deployed to extract specific points, compile them, and find solutions. In this research, GLCM was used to extract seven features to determine malignant and benign nodules.

The type of machine learning and the classifiers used were selected for this research based on the literature. A statistical analysis of the categories of neural networks and their subclassifications that indicated the machine learning technique and classifiers to be used in this study were also based on the literature [5,28,29,30,31,32,33,34,35], as shown in Appendix A, Figure A1.

## 2. Materials and Methods

Based on literature and expert opinion, a systematic, structured approach was adopted to construct an accurate score-based model for thyroid nodule detection.

### 2.1. MATLAB Toolbox

There are numerous toolboxes available such as signal processing, fuzzy logic, neural network, control system, and image processing. For this study, machine learning, image processing, the GUI layout toolbox, and the deep-learning toolbox were used. Many toolboxes were readily available in MATLAB 2018a, and the built-in toolbox was readily available as trial software for the student version.

#### 2.1.1. Image Processing Toolbox

This toolbox provided vital tools for image segmentation, image enhancement, and noise reduction. This pre-image processing and segmentation was a significant part of model building in this study.

#### 2.1.2. Statistics and Machine-Learning Toolbox

The machine-learning toolbox was very helpful in building classification and regression models that drew interpretations of data and used them for training and in predictive models. They provided the applications with the capacity to describe, evaluate, analyze, and build a model. This toolbox was vital because both supervised and unsupervised machine-learning algorithms were provided.

#### 2.1.3. GUI Layout Toolbox

The GUI layout toolbox provided options for displaying the interface from the user’s perspective. MATLAB 2018a was used to build GUI in our study. They provided the applications with the capacity to describe and analyze results from the user’s perspective.

### 2.2. Data Set

A digital database of thyroid ultrasound images was retrieved from an open-source scientific community on 9 February 2019. The primary aim of the open-source data set was for people to use these images in building CAD models for thyroid nodule analyses [36]. It was uploaded and shared online by the Universidad Nacional de Colombia.

An XML file was presented with each image containing the expert’s diagnosis and the relevant patient information. The XML file contained additional information for a better understanding of the characteristics needed for thyroid nodule feature extraction. Patient’s age, sex, nodule composition, shape, margin, calcifications, and TI-RADS score were recorded in .mat format and used for validating results.

The data set images were divided into two categories—benign and malignant. They were also classified under TI-RADS 2, 3, 4a, 4b, 4c, 5. TI-RADS 2 and 3 were considered benign, and TI-RADS, 4a, 4b, 4c, and 5 were considered malignant. In this research, the American College of Radiology TI-RADS classification was used.

A total of 99 patients and 134 ultrasound examinations, were present in the data set. When multiple images were available for some cases, all images were included in order to optimize nodule feature extraction. Among the 99 patients, there were 33 with nodules classified as benign and 66 with nodules classified as malignant.

### 2.3. Image Analysis

The images acquired from the data set had to undergo pre-processing and enhancement (Figure 3). All images had to be resized, changed to the same distance scale, and filtered. Segmentation, which involved the binarization of the image, followed image pre-processing. A threshold was then applied to the image, and further adjustments were made, if required. After this stage, the image was ready for processing.

### 2.4. Image Pre-Processing

A median filter was used in this study. A median filter is a non-linear filter, which, when incorporated, removes impulse noise (salt-and-pepper noise). The median level is determined by the kernel size. Note in Figure 4 and Figure 5 that the filtered image is smoothed, and the high-frequency information of the image is reduced. Boundary pixels seem to be distorted because of the zero-padding effect. Even after omission of the distorted signal, the overall quality of the image in the middle portion remained unchanged [37].

### 2.5. Segmentation

Image segmentation is a process of segregating an image into numerous segments. Segmentation is required to change the representation of an image for easy analysis without altering meaningful information. In this study, our objective for using segmentation was to locate boundaries in thyroid nodule images. Figure 6 and Figure 7 exhibit the segmentation of a filtered image.

Segmentation can be performed with the use of many different methods. In this study, we used the binarization method to convert greyscale images into black and white. The result of OCR (Optical Character Recognition) is black and white, also considered as 0 or 1. Because of the presence of noise in the original image, a high-quality binarized image was used to enhance the OCR result. Furthermore, we used a fixed thresholding method, which used value to assign 0 s and 1 s, and a global binarization method, which used the single value for images [37].

Feature extraction was used to extract data from the visual content of an image, for retrieval and indexing. First-order and second order are the two types of texture feature measures. In this study, a second-order feature measure was used because this measure considers the relationship between neighboring pixels. With the grey level co-occurrence matrix (GLCM), texture feature can be extracted from a given input.

When GLCM is used, the number of rows and columns are equal to the number of quantized grey levels N. Table 1 shows all seven features [38].

### 2.6. Classification

Image classification consists of two phases, namely Training and Testing. After the feature extraction process, the data of the seven characteristics are calculated and separated into classification categories. Based on these, training class is created. Likewise, in the training phase, the separated features are used to classify image features into benign or malignant. A training class is an integral part of the classification process.

In this study, as with all the models that fall under supervised classification, the objective criteria for building a training class were:Independent: Variation of training class data should not impact other values.Discriminatory: Different image features should indicate different characteristics of the thyroid nodule.Reliable: All image features in the training group should share common definitive characteristics with the training class group [39].

Based on the literature survey, dataset, GLCM feature extraction, and training class, four models under supervised learning were built for testing. Two models under classification and two models under regression. After carefully considering factors such as the nature of the data set, results obtained from feature extraction, and training class, the following models were selected:Artificial Neural Network (ANN).Support Vector Machine (SVM) [39].

### 2.7. Performance Metric

In machine learning, evaluating the model by performance measurement is very important. The interpretation of AUC, ROC, accuracy, specificity, and sensitivity is necessary for analysis and in determining if the performance of the model is sufficient or needs further optimization. Each metric or collection of metrics is usually used to check the multi-class classification problem. In this research, the objective was to focus on the accuracy of the model in building computer-aided diagnostics that could classify thyroid nodules as benign or malignant. However, it was also essential to check the performance and efficiency of the model.

Accuracy refers to how well the model can classify nodules correctly. In other words, it evaluates the efficiency and correctness.
$$Acc=\frac{TP+TN}{TP+TN+FP+FN}$$
where True Positive (*TP*). False Positive (*FP*). True Negative (*TN*). False Negative (*FN*).

Sensitivity is also called the hit rate or recall. It measures the proportion of the total positive samples correctly identified by the model.

It is usually calculated for the probability of 1, whereas specificity, also called an inverse recall, is the proportion of all negative samples correctly classified as negative. Sensitivity and specificity act as two kinds of accuracy. Sensitivity is for actual positive samples and specificity is for actual negative samples. Therefore, both can be used for evaluating model performance.
$$TPR=\frac{TP}{TP+FN}=\frac{TP}{P},\;TNR=\frac{TN}{FP+TN}=\frac{TN}{N}$$
where Sensitivity = True Positive Rate (*TPR*), Specificity = True Negative Rate (*TNR*).

The *F*1-*score* is the harmonic mean of *Precision* and *Recall* and gives a better measure of the incorrectly classified cases than the Accuracy Metric.
$$F1\;Score={\left(\frac{Recall^{-1}+Precision^{-1}}{2}\right)}^{-1}=2*\frac{\left(Precision*Recall\right)}{\left(Precison+Recall\right)}$$

### 2.8. Artificial Neural Network (ANN)

The neural network model was perceived as a mathematical model, which defines a function, as shown by the following formula:f:X→Y

According to the literature, a neural network is commonly known as ANN. It is a definition of a class in functions where each node of a class is obtained by changing parameters, connection weights or architecture (number of neurons), or connectivity [40].

In a neural network, the following formula is used:$$f\left(x\right)\to gi\left(x\right)$$
where *f(x)* is the compositions of other functions *gi(x)*; *gi(x)* can be disintegrated into other functions.

Nonlinear weighted sum uses the following formula:$$f\left(x\right)=K\left(\sum i)wigi\left(x\right)\right)$$
where *K* can be a hyperbolic tangent, a sigmoid function, a softmax function, or a rectifier function.

This is also known as the activation function, which provides a smooth transition for any input variance. Collection of a function is represented as follows:gi →Vector g
$$g=\left(g1,g2,g3\ldots gn\right)$$

Figure 8 shows the disintegration of h1; the arrows specify variables between dependencies. This can be viewed as follows:

View 1: 3D vector h transformed from input x. 3D vector h further transformed into 2D vector g, and finally transformed into f.

View 2: Random variable F = f(G) depends on G = g(H), and further depends on H = h(X).

The architecture of View 1 and View 2 and its components of layers are independent of one another. Because of the directed acyclic graph, such a network is called feedforward artificial neural network [41]. Figure 9 shows multilayer ANN. Appendix B describes Support Vector Machine-SVM.

## 3. Results

### 3.1. Image Analysis and Image Processing

Figure 10a–c shows the image pre-processing and segmentation of the original image of a benign nodule. The original image is resized into 256 × 256 mm. Figure 10b shows a filtered image. The variation between the original image (Figure 10a) and the median filtered image (Figure 10b) is evident. The filtered image is smoothed; the high-frequency information of the image is reduced. Boundary pixels appear to be distorted because of the zero-padding effect. Figure 10c shows a binary segmented image that locates the boundaries of thyroid nodules. In this segmented image, there is a gap in the center pointing out the nodule for feature extraction. This was classified as a benign image.

Figure 11a–c shows the image pre-processing and segmentation of the original image of a malignant nodule. The process of image pre-processing and segmentation is the same as that used for a benign nodule. In the images with multiple nodules (such as Figure 11a,b, which shows two nodules next to one another in the center), the area is pointed out for feature extraction. This image was classified as a malignant nodule.

### 3.2. Accuracy-Based Predictive Model and Optimization

The predictive analysis for the training class was divided into three categories: benign, malignant and test (includes benign, malignant, and additional images). As this study was based on the accuracy score and the model’s stability, over-optimization was avoided. Two models, ANN and SVM, achieved accuracy above 70% in the test prediction and are mentioned in this section.

Predictive models for thyroid nodule detection with good accuracy score, sensitivity, and specificity. ANN and SVM produced a decent accuracy score. The accuracy of the SVM model was 96%, and the accuracy of the ANN model was 75%.

Sensitivity describes how often a model correctly predicts a positive result for people with the condition that is tested for. For ANN, the sensitivity is 0.4914, and for SVM, it is 0.7866. Therefore, SVM outperforms ANN in sensitivity. Specificity describes how often a model correctly predicts a negative result for people without the condition tested for. Both ANN and SVM have a high specificity of 1.

Table 2 specifies the performance of both the model. The F1 score represents the harmonic mean of recall, and precision ranges from 0 to 1. The value indicates high-classification performance and the results demonstrated that SVM outperformed ANN. Table 3 shows the Confusion matrix for SVM test phase.

### 3.3. Graphic User Interface (GUI) from the Support Vector Machine (SVM)

After closely examining both ANN and SVM performance, it was decided that a Graphic User Interface would be designed for SVM. As SVM performed better in all parameters with high accuracy, sensitivity, and specificity, feature classification was obtained from the SVM model analysis as it had the best accuracy at 96%.

After running the GUI code, this window opens and prompts image selection from the test images file.

The images are uploaded to the input image window as shown in Figure 12; as soon as the SVM code is completed, the GUI displays the features.

The GUI was designed to display Nodule Composition, Echogenicity, Margin, Classification, and Type as shown in the Figure 13. For a highly suspicious model, a window prompt was used to recommend an FNAB to radiologists and doctors as shown in Figure 14. Four features for the GUI display were selected based on expert opinion. These four features demonstrated a high TPR in the SVM validation phase. Therefore, we chose these features in the GUI in order to give the clinician the most important information.

## 4. Discussion

This research described the ultrasound classification of thyroid nodules using machine learning techniques. An important aspect of this study is the completion of pre-processing with the use of median filters as the median filter is non-linear and can preserve essential details even after omitting noise. In this process, it was noted that median filters were advantageous over mean filters because no neighboring pixel would vary the median value.

Moreover, edge preservation is vital, and median filters did not add new pixels in the image. We found a significant reduction of noise with a change in contrast as compared to the original image. To divide an image into many segments, the segmentation process was used. This process located the boundary of the nodule and indicated the area of interest for feature extraction. Optical Character Recognition (ORC) enhanced the image by reducing the high-frequency noise and assigned values for 0 and 1 (black and white), with a fixed threshold.

In this study, two models were constructed. SVM outperformed ANN on all performance metrics, achieving an accuracy score of nearly 96%.

Regarding the construction of the model, SVMs perform well under a changing threshold with the introduction of the flexible kernel. It is non-linear and non-parametric, supporting different functions from all classifications of data. SVMs can provide out-of-sample generalization. This is due to SVM being robust even when there is bias in the training class data. In a way, SVM outperforms neural networks since the optimality problem is convex. A neural network has multiple solutions associated with local minima.

In this study we took a slightly different approach. According to our literature, we recognized that only 2% of all CAD use ANN with supervised learning. Therefore, we wanted to explore a highly accurate and stable model with minimum optimization.

There is only one published study to compare our results with [29]. However, they used different models such as 10-fold validation and optimization, a larger data set, and different GLCM texture features for image processing. Table 4 shows the comparison ROC curve parameters.

The optimization of SVM and ANN was vital, varying the specific parameters in the training class to obtain better performance from the classifier. To use data effectively, the training class used most of the images as compared to the test class. A ratio of 87:13 was used for the training and testing classes for both SVM and ANN. A ratio of 85:15 was also tried, and ANN’ accuracy decreased by a significant margin. For SVM variables, constraints and objective functions used were taken from libraries.

Clinically, our model performance suggests that it could reliably distinguish benign from malignant nodules. It has been suggested that the CAD systems could be used as a first reader or a second reader to the radiologist, or even as an autonomous reader. Further development of CAD systems such as those presented in our study would require careful planning as to how they could be integrated into clinical practice workflows. Validation studies of integrated CAD systems would be required for real-world application to become evidence based. There is also clinical uncertainty as to the best way to apply CAD models—whether their purpose should be to have a high positive predictive value (i.e., if positive, the nodule is likely to be malignant and an FNAB would be warranted) or to have a high negative predictive value (i.e., if negative, the nodule is not likely to be malignant and an FNAB would not be warranted). If the radiologist is less experienced, a model that has a high negative predictive value may be preferred. Significant consideration should also be given to how integrated CAD models will influence the process of shared decision making between clinicians and patients regarding the management of thyroid nodules.

The successful implementation of CAD systems in real-world clinical practice could potentially reduce time spent by radiologists assessing ultrasounds, reduce the need for FNABs, and reduce interreader variability associated with TI-RADs. Recent studies of deep learning neural algorithms to provide recommendations for thyroid nodule biopsy are promising.

Future research in this area should use much more substantial datasets, and ones that come from two ultrasound sources. Future research should build two models—one under supervised learning and another under unsupervised learning—and try to integrate both models and compare performance metrics. Reinforcement machine learning algorithms should be deployed in future research because they are capable of choosing an action, building on each data point, and reviewing its own decision. These algorithms re-learn from their shortcomings and produce enhanced results each time. Platforms like Python or R should be used instead of MATLAB, as in this research. More information should be extracted to assist radiologists, which may be applied as a performance metric in future research and compared to the performance of the CAD system.

## 5. Conclusions

This research achieved its aim of building a CAD model that could classify thyroid nodules as benign or malignant, and the aim of evaluating its performance, demonstrating that an SVM model has superior performance as compared to an ANN model. This work demonstrates the significance of using traditional ML approaches with minimum optimization, stability, and successful CAD systems.

## Figures and Tables

**Figure 1 medicina-57-00527-f001:**
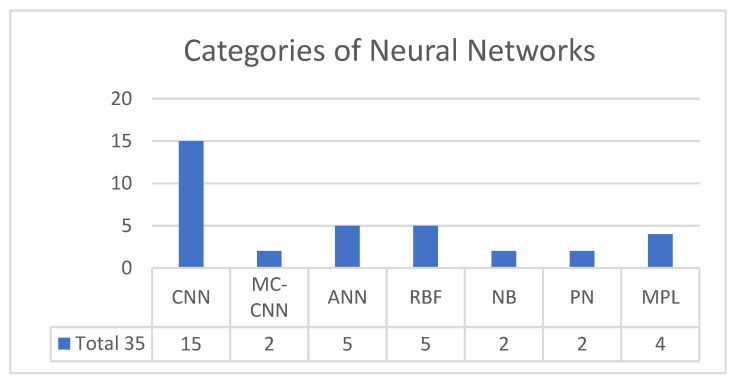
Categories of neural network for automated thyroid nodule detection.

**Figure 2 medicina-57-00527-f002:**
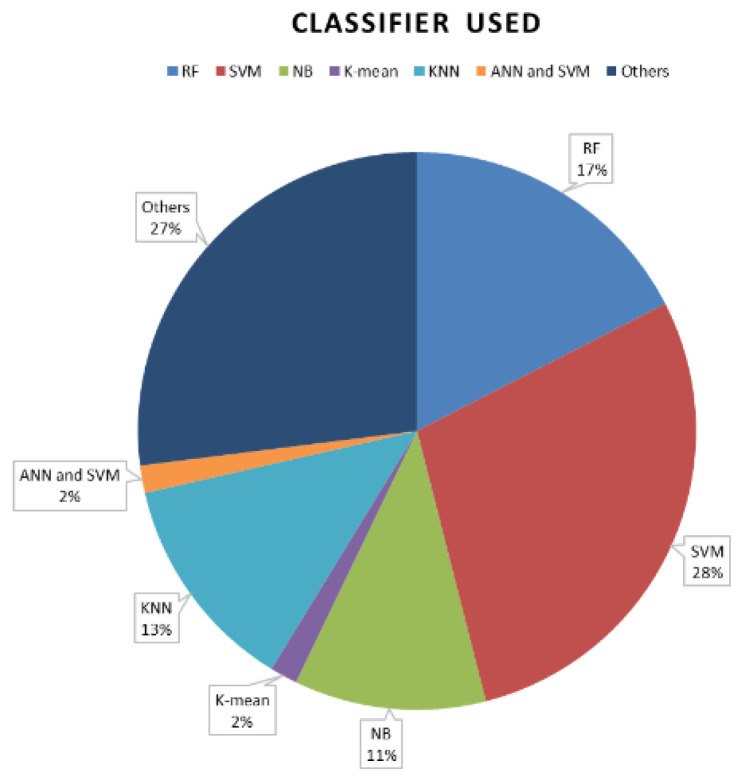
Classifier used.

**Figure 3 medicina-57-00527-f003:**
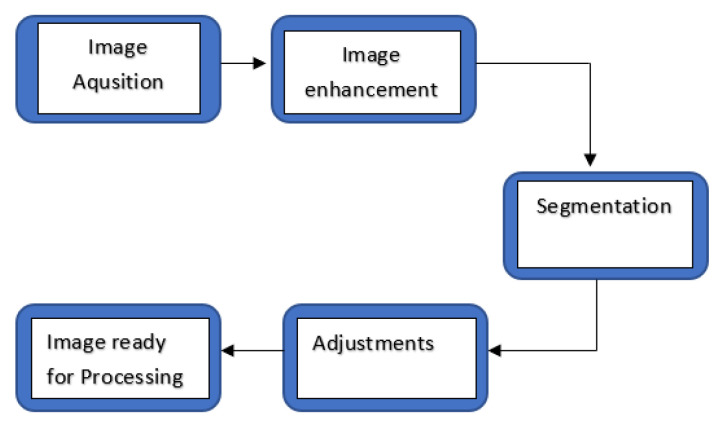
Image processing flow chart.

**Figure 4 medicina-57-00527-f004:**
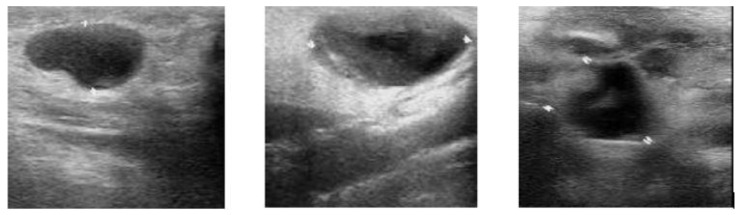
Original ultrasound scanning image.

**Figure 5 medicina-57-00527-f005:**
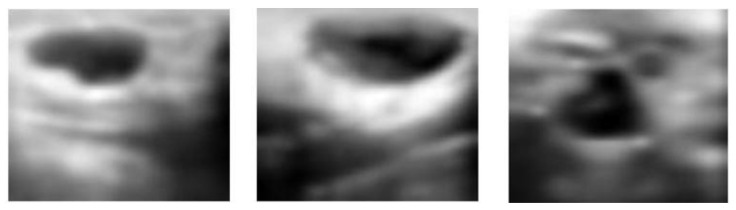
Filtered image.

**Figure 6 medicina-57-00527-f006:**
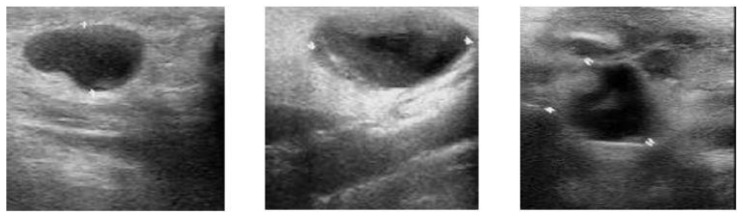
Original Ultrasound scanning Image.

**Figure 7 medicina-57-00527-f007:**
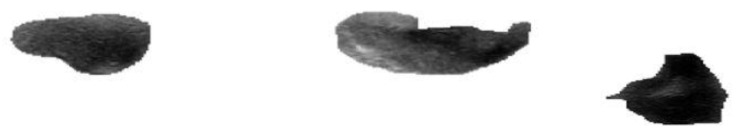
Segmentation process output.

**Figure 8 medicina-57-00527-f008:**
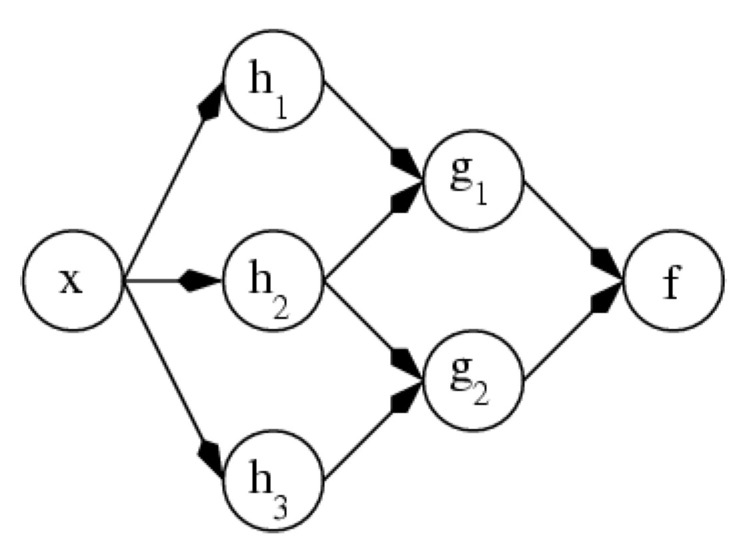
Feedforward artificial neural network [40].

**Figure 9 medicina-57-00527-f009:**
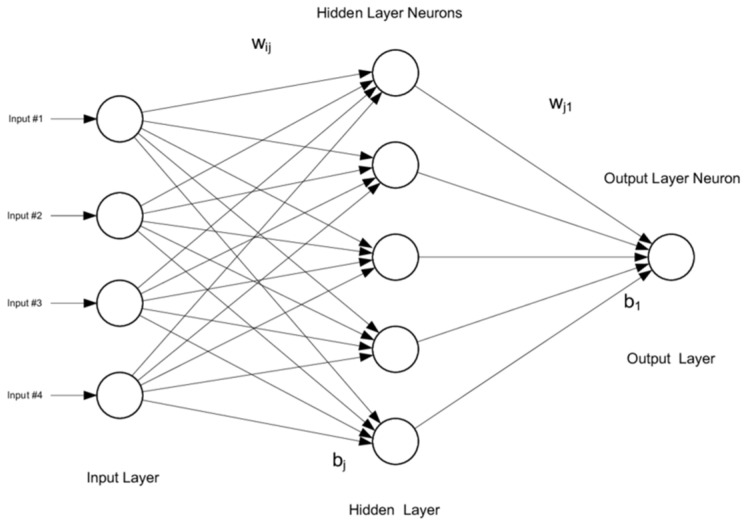
Multilayer artificial neural network [42].

**Figure 10 medicina-57-00527-f010:**
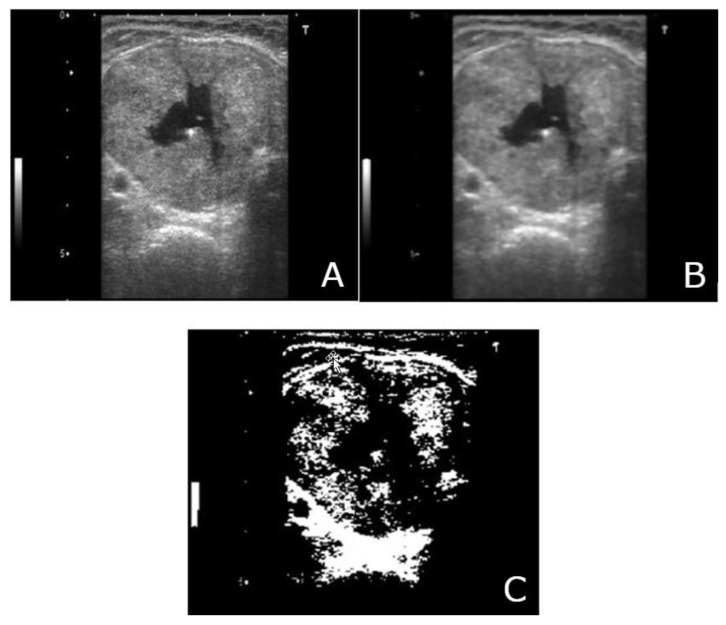
(**A**) Original image (benign)-Top Left; (**B**) median filtered image (benign)-Top Right; (**C**) segmentation-Center Bottom.

**Figure 11 medicina-57-00527-f011:**
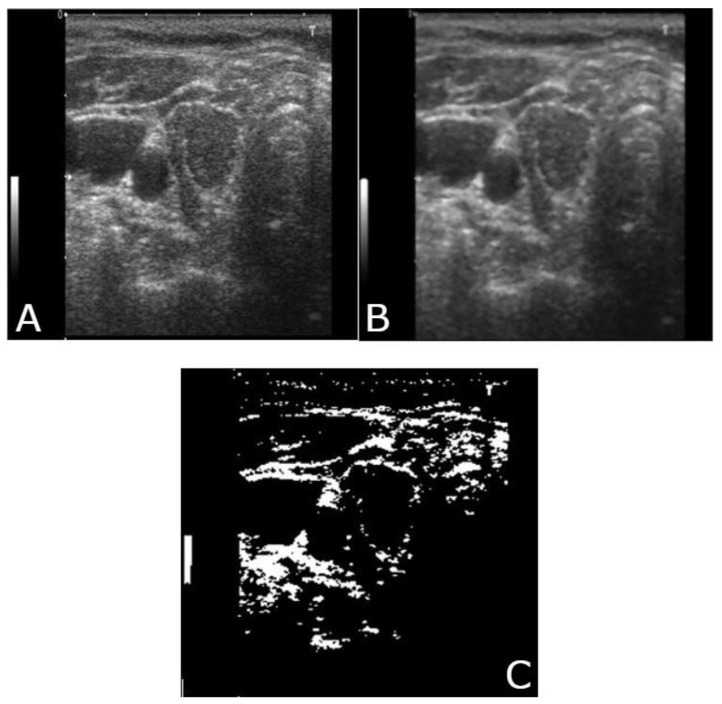
(**A**) Original image (malignant)-Top Left; (**B**) median filtered image (malignant) Top Right; (**C**) binary segmentation-Center Bottom.

**Figure 12 medicina-57-00527-f012:**
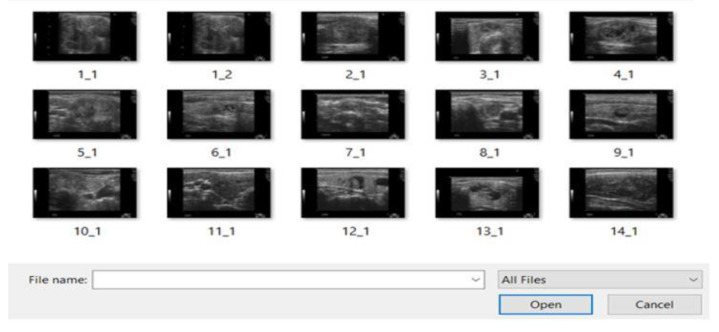
GUI prompt.

**Figure 13 medicina-57-00527-f013:**
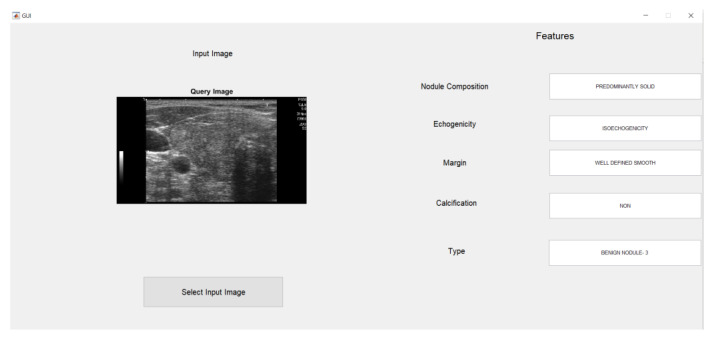
GUI for benign nodules.

**Figure 14 medicina-57-00527-f014:**
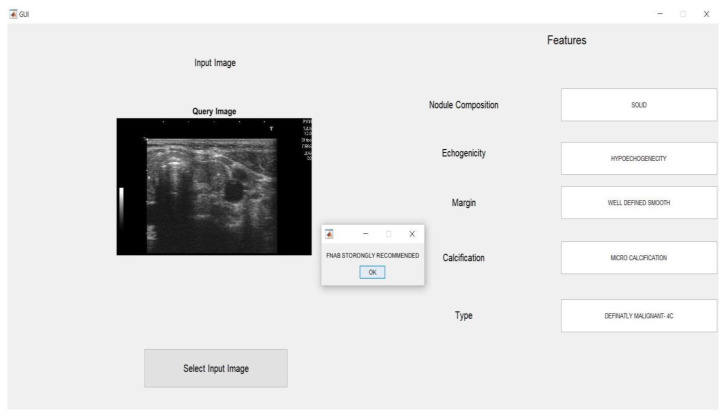
GUI for malignant nodule.

**Table 1 medicina-57-00527-t001:** Feature Extraction Formulas.

Features	Equations
Energy	$\overset{N-1}{\underset{i=0}{\sum }}\overset{N-1}{\underset{j=0}{\sum }}p{\left(i,j\right)}^{2}$
Correlation	$\overset{N-1}{\underset{i=0}{\sum }}\overset{N-1}{\underset{j=0}{\sum }}\frac{\left(i-\mu i\right)\left(j-\mu j\right)p\left(i,j\right)}{\sigma_{i}\sigma_{j}}$
Entropy	$-\overset{N-1}{\underset{i=0}{\sum }}\overset{n-1}{\underset{j=0}{\sum }}p\left(i,j\right)log\left(p\left(i,j\right)\right)$
Homogeneity	$\overset{N-1}{\underset{i=0}{\sum }}\overset{N-1}{\underset{j=0}{\sum }}\frac{p\left(i,j\right)}{1+\left|i-j\right|}$
Cluster Shade	$\overset{N-1}{\underset{i=0}{\sum }}\overset{N-1}{\underset{j=0}{\sum }}{\left(i+j-\mu_{x}-\mu_{y}\right)}^{3}p\left(i,j\right)$
Contrast	$\overset{N-1}{\underset{i=0}{\sum }}\overset{N-1}{\underset{j=0}{\sum }}{\left|i-j\right|}^{2}p\left(i,j\right)$
Inverse Difference Movement	∑i=0N−1∑j=0N−111+i−jpi,j

**Table 2 medicina-57-00527-t002:** Performance metric of ANN and SVM test phase.

ROC Curve Parameters	ANN—Test Phase	SVM
Distance	0.5086	0.7416
Threshold	0.0078	16
Sensitivity	0.4914	0.7866
Specificity	1	1
Accuracy	74.5684	96
PPV	1	0.7857
FNR	0.5086	0.2134
FPR	0	0
*F*1 *score*	0.659	0.92

**Table 3 medicina-57-00527-t003:** Confusion Matrix for SVM test phase.

Actual	Predicted		Accuracy = 0.96
	Positive	Negative	
Positive	*TP* = 11	*FN* = 3	P = 14
Negative	*FP* = 0	*TN* = 62	N = 62
	*PP* = 11	*PN* = 65	M = 76

**Table 4 medicina-57-00527-t004:** Performance metric comparison from literature (ANN) perspective.

ROC Curve Parameters	This Study	Previous Study
Supervised Learning	Classification	Classification
Classification	ANN	Decision tree, random forest model
Secondary Model		
Sensitivity	0.4914	0.789
Specificity	1	0.785
AROC	0.7475	0.84
Accuracy	74.5684	78.5
PPV	1	0.785
FNR	0.5086	Not Specified
FPR	0	0.215
*F*1 *score*	0.659	0.784

## Data Availability

In Accordance with MDPI Research Data Policies.

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
