# Peer review of "Ultrasound Image Classification of Thyroid Nodules Using Machine Learning Techniques"

_medicina, 2021, doi:10.3390/medicina57060527_

Round 1
Reviewer 1 Report
In this article, the author developed a computer-aided diagnosis system integrated with multiple-instance learning (MIL) to focus on benign-malignant classification. They also developed a graphic user interface (GUI) to display the image features to help radiologists’ decision-making.
The paper has a proper structure, readability and length. Minor details on English grammar require review. Also, the novel contribution of the paper is highlighted, as well. I think this article has good potential but, before being considered ready for publication, some aspects need to be clarified and improved.
1)Please add a separate section entitled "Related Work" to transit smoothly in the following parts of the work. My suggestion is to increase the number of references to allow for deeper discussion and comparisons with the state-of-the-art.
2)The "Materials and Methods" section needs enrichment. The model needs to be explained more technically with more details from ML technical points of view. The authors need to elaborate, why the traditional ML classifiers are selected. Technical reasons must provide in contrast to the state-of-the-art Machine learning approaches. The authors did not explain enough how these methods used. Please, explain them clearly.
3)The presented methodology and the results are not communicated, with the necessary background for the readers included in the paper. Without the experimental description, the results are doubtful. Please, demonstrate the environment of the experiment in detail. There is no clear presentation of the results and their commentary. Try to make a more coherent, accurate and focused presentation.
4)The discussion part needs to be more technical and also highlight future works. I recommend the authors discuss the merits of the proposed approach. Before Conclusion, I suggest the authors draw a Table and compare it with previous researchers, how your approach is better in terms of accuracy.
5)The conclusion section could be further improved to highlight the proposed results. Moreover, conclusions can discuss future research directions and extensions of the study.
6)I recommend the authors include more recent references and from MDPI as well.
It is clear the overall merit and contribution of this work.
In my opinion, the work should be enriched with up-to-date literature and emphasize the application nature of the solution's use.
Author Response
School of Medicine,
National University of Ireland Galway,
Galway,
Ireland
Editor,
Medicina
Date typed: May 10th, 2021,
Dear Editor,
Thank you for considering out manuscript. We found the reviewer comments helpful and constructive. Below we outline our response to each of the comments and the relevant changes made to the manuscript. if you find these satisfactory , please consider the manuscript further for publication in your journal.
1)Please add a separate section entitled "Related Work" to transit smoothly in the following parts of the work. My suggestion is to increase the number of references to allow for deeper discussion and comparisons with the state-of-the-art.
Answer: As suggested we have added a separate section named " Previous Related Work", and Annex 1. We have clearly explained our Literature search strategy, decision making and technical consideration of our approach. We have added number of references from our literature search.
Previous Related Work.
For this study, a comprehensive literature review which included 35 research papers on ultrasound thyroid nodule with machine learning system was completed. Numerous CAD system has been researched for automated thyroid detection in recent years. In our literature review, we found that most CAD systems are based on Convolution Neural Network (CNN) Radial basis function (RBF) neural network which used K-NN for high accuracy scores. Moreover, multitask cascade convolution neural network (MC-CNN) requires a more extensive data set for a training class. (14), (15), (16), (17), (18) (19) (20)
Since there was already a lot of research papers on CNN, we wanted to deploy different approach. Therefore, we selected Artificial neural network (ANN) as our primary model. (Figure 1) shows the number of research papers from our literature studies uses various approach. Probabilistic neural networks (PN), Multilayer Perceptron Neural Network (MLPNN). (21), (22), (23), (24), (25), (26), (27), (28), (29).
Figure 1. Categories of neural network for automated thyroid nodule detection
Figure 2. Classifier Used
Selection of classifiers mainly depends on how well the model is trained and tested. (Figure 2) shows the number of research papers from our literature studies uses various approach. SVM is widely used because of its flexible kernel function, and threshold. 28% of the research conducted on automated thyroid is based on SVM algorithm. However, only 2% of the research conducted was based on ANN and SVM, Therefore, we opted for SVM approach.
For image processing and segmentation, median filters have numerous advantages over mean filters and Binarization Method to convert greyscale images into black and white this approach was workable and straightforward. Therefore, these methods were adopted. In computer vision, for any complicated computational task, feature extraction is deployed to extract specific points, compile it, and find solutions. In this study, GLCM was used to extract seven features to determine malignant and benign nodule. The choice of machine learning technique and selection of classifiers was mainly based on a literature review. Statistical analysis for categories of neural network and their subclassification indicated machine learning technique and classifiers to be used in this study. (30), (31), (32), (33), (34) (35).
Annex 1.
Most of the papers in the literature review used Supervised learning due to the nature of data. With the feedback loop and using both input and output data for the predictive model guarantees high accuracy, however, using supervised learning and further optimizing it by unsupervised learning models has produced high accuracy score. Unsupervised models are specificity-based models which yield low accuracy for the size of our data set. Figure 18 shows the number of research papers from our literature studies uses various approach.
Accuracy, Sensitivity and Specificity based models adopt the classification method were as Mean Squared Error, Mean Absolute Error based models adopt regression approach. A combined approach yields stable models. Figure 19 shows the number of research papers from our literature studies uses various approach.
In computer vision, for any complicated computational task, feature extraction is deployed to extract specific points, compile it and find solutions. In this thesis, GLCM was used to extract seven features to determine malignant and benign nodule.
2)The "Materials and Methods" section needs enrichment. The model needs to be explained more technically with more details from ML technical points of view. The authors need to elaborate, why the traditional ML classifiers are selected. Technical reasons must provide in contrast to the state-of-the-art Machine learning approaches. The authors did not explain enough how these methods used. Please, explain them clearly.
Answer: This point is addressed in our answer to comment 1. Please refer to Comment 1, Answer and Annex 1- why the traditional ML classifiers are selected. Technical reasons must provide in contrast to the state-of-the-art Machine learning approaches.
We have enhanced "Materials and Methods" section by including details on MATLAB Toolbox, performance metric with formulas used. We have also included Technical details of ANN as it is our primary model. In Annex 2 we have details of SVM Technical details.
MATLAB Toolbox.
In this study we have used MATLAB Toolbox for our analysis.
There is numerous toolbox available such as signal processing, fuzzy logic, neural network, control system and image processing. For this study, Machin learning, Image processing, GUI layout toolbox and deep learning toolbox. Many toolboxes were readily available in MATLAB 2018a, and the built-in toolbox was readily available for trail software for student’s version.
Image Processing Toolbox: -
This toolbox provides vital tools to perform image segmentation, image enhancement, noise reduction. This pre-image processing and segmentation is a significant part of a model building in this study.
Statistics and Machine Learning Toolbox: -
Machin learning toolbox is very helpful for building classification and regression models to draw interpretations from data and use it to train and predictive models. They provide the function with applications to describe, evaluate, analyse, and build a model. This toolbox was vital because both supervised and unsupervised machine learning algorithm are provided.
Performance Metric: -
In machine learning, evaluating the model by performance measurement is very important. Interpretation of AUC, ROC, accuracy, specificity, and sensitivity is necessary to analyse and determine if the performance of the model is sufficient or needs further optimization. Each metric or collection of metric is usually used to check the multi-class classification problem. In our study, we are more focused on the accuracy of the model as our main objective is to build a Computer-aided diagnostics to classify benign or malignant thyroid nodules. However, it is also essential to check the performance and efficiency of the model.
Accuracy
Accuracy is the percentage specifying how well model is capable of separating into its categories. In other words, it evaluates the efficiency and correctness.
Were, True positive (TP).
False Positive (FP).
True Negative (TN).
False Negative (FN).
Sensitivity and Specificity
Sensitivity is also called the hit rate or recall. It is the measure for positive correctly classified samples to total positive samples. Usually, it is calculated for the probability of 1. Wereas specificity, also called an inverse recall. It is the measure of correctly classified negative samples to the total number of negative samples. Sensitivity and specificity act as two kinds of accuracy. Sensitivity for actual positive samples and specificity for actual negative samples. Therefore, both can be used for evaluating model performance.
Where, Sensitivity = True positive rate (TPR).
Specificity = True Negative rate (TNR).
Area Under the Receiver Operating Characteristics (AUROC)
AUC- ROC curve is a performance metric used for numerous thresholds settings. AUC presents the degree or measure of separability. Based on the literature survey AUC- ROC is one of the best metric to evaluate the performance of the model. It represents capability between classes. Therefore, it is interpreted that higher the AUC value, better the model prediction of benign as benign and malignant as malignant.
Artificial Neural Network (ANN) .
Neural network model was perceived as a mathematical model, which defines a function.
f:X→Y
From literature, A neural network is commonly known as Artificial network (ANN). It is a definition of a class in functions where each node of a class is obtained by changing parameters, connection weights or architecture (number of neurons) or connectivity.
In a neuron network,
f(x) gi(x)
where, f(x) is compositions of other functions gi(x).
gi(x) can be disintegrated to other functions.
nonlinear weighted sum
{\displaystyle \textstyle f(x)=K\left(\sum _{i}w_{i}g_{i}(x)\right)}f(x)=K(
where, K can be a hyperbolic tangent, sigmoid function, softmax function, or rectifier function.
For our analysis, we have used activation function, which provides a smooth transition for any input variance. Collection of a function.
gi as a vector {\displaystyle \textstyle g=(g_{1},g_{2},\ldots ,g_{n})}g=(g1,g2,g3,...,gn).
Figure 9 shows the disintegration of h1; the arrows specify variables between dependencies. This can be viewed as.
Figure 9. Feedforward Artificial neural network
View 1: 3D vector h transformed from input x. 3D vector h further transformed into 2 D vector g and lastly transformed into f.
View 2: Random variable F=f(G) depends on G=g(H) further depends on upon H=h(X),
We have incorporated architecture of View 1 and View 2 and its components of layers are independent of one and other. Because of the directed acyclic graph, such a network is called feedforward Artificial neural network. (36)
Figure 5. Multilayer Artificial Neural Network.
Annex 2
Support Vector Machine (SVM)
Support vector machine (SVM) was the method used to accomplish the classification of benign and malignant by constructing hyperplane in a multidimensional space. They later separate different class labels. In our study, SVM was used to support classification tasks under supervised learning. SVM creates a dummy variable 0 or 1. Therefore; the dependent variable consists of three variables. Dummy variables are represented as X, Y, Z
X: {1 0 0}, Y: {0 1 0}, Z: {0 0 1}
To negate the error function SVM uses iterative training algorithm, to build an optimal hyperplane.
Because we were using a classification model in our study, SVM models are categorized in two. They are as follows.
- Type 1 SVM Classification(C-SVM).
- Type 2 SVM Classification (nu- SVM).
Type 1 SVM Classification(C-SVM): -
To negate the error function, we used
Constraints are.
Where C capacity constant.
w vector of coefficients.
b constant
nonseparable data (inputs)
The index I tags the N, which is training cases. Represents the class labels, and xi is independent variables. I was the kernel for the transformation of data from input to feature. (40)
Type 2 SVM Classification (nu- SVM) : -
To negate the error function, we used
Constraint is.
In type 2 SVM, there was a need to estimate the functional dependence of variable y on independent variables x. The relationship between x and y is by deterministic function f plus.
SVM is supported by many kernels such as linear, polynomial, radial basis function (RBF) and sigmoid.
Were,
A dot product of input data points is mapped into a higher dimensional feature by
3)The presented methodology and the results are not communicated, with the necessary background for the readers included in the paper. Without the experimental description, the results are doubtful. Please, demonstrate the environment of the experiment in detail. There is no clear presentation of the results and their commentary. Try to make a more coherent, accurate and focused presentation.
Answer: We have enhanced results section by including description on predictive models and optimization and Graphic user interface. Since our objective of the study is accuracy-based model with Stability, we have avoided over-optimization. In Annex 3 we have included results of Validation phase for both benign nodules and Malignant nodules.
3.2. Accuracy based predictive model and optimisation.
The predictive analysis for training class was divided into three categories benign, malignant and Test (includes benign, malignant, and additional images). This approach was adopted to understand closely True positive rate (TPR) and true negative rate (TNR). TPR and TNR were used as the first indicator for the efficiency of the model. Further based on these values’ threshold was adjusted to obtain good accuracy. As this study was based on accuracy score and stable model, over-optimization was avoided. Two models, ANN and SVM, achieved accuracy above 70% in the test prediction and are mentioned in this section.
|
ROC curve Parameters. |
ANN |
SVM |
|
Distance |
0.5086 |
0.7416 |
|
Threshold |
0.0078 |
16 |
|
Sensitivity |
0.4511 |
0.49 |
|
Specificity |
1 |
1 |
|
AROC |
0.7475 |
1 |
|
Accuracy |
74.5684 |
89.8739 |
|
PPV |
1 |
0.8965 |
|
FNR |
0.5086 |
0.2134 |
|
FPR |
0 |
0 |
|
F1 score |
0.659 |
0.989 |
3.3. Graphic User Interface (GUI) from the Support Vector Machine (SVM).
After closely examining both ANN’s and SVM’s performance, it was decided to design Graphic User Interface for SVM. As SVM performed better in all parameters with high accuracy, sensitivity and specificity, features classification was obtained from SVM models analysis as it had the best accuracy of 90%.
After running the GUI code, this window opens and prompts for any image selection from the test images file.
Figure 15. GUI Prompt.
The images are uploaded in the input image window, as soon as SVM code is completed, the GUI displays features.
Figure 16. Benign Nodule
Figure 17. Malignant Nodule
GUI was designed to display Nodule Composition, Echogenicity, Margin, Classification and Type. For highly suspicious model a windows prompt was used to recommend radiologists and doctor for FNAB.
Annex 3: Validation phase results.
Artificial Neural Network (ANN) Benign nodules Validation.
Artificial Neural Network (ANN) Malignant nodules Validation.
|
ROC curve Parameters. |
ANN(Benign)- Validation phase. |
ANN(Malignant)-Validation phase. |
ANN- Test phase |
|
Distance |
0.1437 |
0.5489 |
0.5086 |
|
Threshold |
0.0078 |
0.0078 |
0.0078 |
|
Sensitivity |
0.8563 |
0.4511 |
0.4511 |
|
Specificity |
1 |
1 |
1 |
|
AROC |
0.9281 |
0.7255 |
0.7475 |
|
Accuracy |
92.8148 |
72.5549 |
74.5684 |
|
PPV |
1 |
1 |
1 |
|
FNR |
0.1437 |
0.5489 |
0.5086 |
|
FPR |
0 |
0 |
0 |
|
F1 score |
0.9226 |
0.6217 |
0.659 |
Support Vector Machine SVM Validation Phase
|
ROC curve Parameters. |
SVM Validation |
|
Distance |
0.7416 |
|
Threshold |
16 |
|
Sensitivity |
0.94 |
|
Specificity |
0.97 |
|
AROC |
1 |
|
Accuracy |
91.8739 |
|
PPV |
0.8965 |
|
FNR |
0.2134 |
|
FPR |
0 |
|
F1 score |
0.989 |
4)The discussion part needs to be more technical and highlight future works. I recommend the authors discuss the merits of the proposed approach. Before Conclusion, I suggest the authors draw a Table and compare it with previous researchers, how your approach is better in terms of accuracy.
We have enhanced discussion section by adding key technical information, Future Works, As suggested we have compared our findings with previous study.
Important aspects of this study were that pre-processing was completed using median filters because the median filter is non-linear and can preserve essential details even after omitting noise. In this process, it was noted that median filters are advantageous over mean filters because no neighboring pixel will vary the median value.
Moreover, edge preservation is vital, and median filters did not add new pixels in the image. We found significant reduction of noise with a change in contrast to the original image. To divide an image into many segments, the segmentation process was used. This was to locate the boundary of the nodule and indicate the area of interest for feature extraction Optical Character Recognition (ORC) enhanced the image reducing the high-frequency noise and assign values for 0 and 1 (Black and white) concerning a fixed threshold.
In this study we have taken slightly different approach, from our literature we recognized only 2% of all CAD use ANN with supervised learning. Therefore, we wanted to explore high accuracy stable model with minimum optimization.
There is only one recognized study to compare our results with (29). But they have used different models, 10-fold validation & optimization, a larger data set and different GLCM texture features for image processing.
Table shows comparison ROC curve parameters.
|
ROC curve Parameters. |
This Study |
Similar Study (29) |
|
Supervised Learning |
Classification |
Classification |
|
Classification |
SVM |
Decision tree, random forest model |
|
Sensitivity |
0.4914 |
0.789 |
|
Specificity |
1 |
0.785 |
|
AROC |
0.7475 |
0.84 |
|
Accuracy |
74.5684 |
78.5 |
|
PPV |
1 |
0.785 |
|
FNR |
0.5086 |
Not Specified |
|
FPR |
0 |
0.215 |
|
F1 score |
0.659 |
0.784 |
Optimization of SVM and ANN was vital, varying the specific parameters in training class to obtain better performance of the classifier. To use data effectively, the training class used most of the image compared to the test class. 87: 13 ratios were used for training and testing class for both SVM and ANN. 85:15 ratio was also tried, and ANNs accuracy decreased by a significant margin. For SVM variables, constraints and objective functions were used from libraries.
Our main objective of this study was to avoid over optimization, we believe our predictive model is more stable. They have achieved better accuracy, but it is unclear if the model is stable. Although, similar study’s false positive rate is low (0.215). We have achieved false positive rate of 0 demonstrating stable model. Our model has specificity 1 with invers recall measure of correctly classifying negative samples to the total number of negative samples. In malignant nodule detection Sensitivity and Specificity score indicates performance of CAD. They have not specified what is their accuracy of secondary model which is used for classification.
5)The conclusion section could be further improved to highlight the proposed results. Moreover, conclusions can discuss future research directions and extensions of the study.
Answer: We have outlined key successful achievements of this study and based on our experiences; we have suggested further recommendations for researchers.
This research has achieved its aim to build a CAD model to classify thyroid nodules as benign or malignant and evaluate its performance, demonstrating that an SVM model has a superior performance to an ANN model. This work demonstrates the significance of using traditional ML approaches with minimum optimization, stability, and successful CAD systems.
Recommendations for other researchers who may try a similar approach in the future.
- Collect a much more substantial dataset, five times the size used in this study. From two ultrasound scanners.
- Build two models, each under supervised learning and unsupervised learning, and try to integrate both models and compare performance metrics.
- Deploy Reinforcement Machine Learning Algorithms; they can choose an action, built on each data point and review its own decision. The algorithm re-learns from its shortcomings and produces enhanced results each time.
- Use a platform like Python or R instead of MATLAB.
- Use radiologists as a performance metric and compare results with the CAD system. Deploy tenfold validation for the ANN model.
6)I recommend the authors include more recent references and from MDPI as well. It is clear the overall merit and contribution of this work.
In my opinion, the work should be enriched with up-to-date literature and emphasize the application nature of the solution's use.
Answer: We have expanded our scope of Literature till 2021. We have reviewed articles from MDPI and added references.
We hope you find these responses to the reviewer comments satisfactory and look forward to hearing your response.
Yours Sincerely,
Vijay Vyas Vadhiraj,
School of Medicine,
National University of Ireland Galway

Reviewer 2 Report
This manuscript by Vadhiraj et al. studies the possible role of Machine learning techniques as an auxiliary diagnostic tool in distinguishing benign and malignant thyroid nodules on a US base. The topic is of clinical relevance, but I have several concerns mainly related to the methods proposed and the results described, limiting the possibilities to draw any conclusions based on the data reported in this manuscript. Particularly:
Introduction
General comment: the background of the study has to be improved, and additional relevant references added: Computer-aided diagnostic system and Artificial Intelligence algorithm in thyroid diagnostics is a fascinating topic and has to be properly introduced, and the manuscript contextualized accordingly.
- Page 2, second paragraph, lines 15-17: the US reporting system referred by authors is the TI-RADS. However, TI-RADS is a generic term that refers to several US reporting systems, each with specific diagnostic classes, risk of malignancies, and clinical recommendations. The TI-RADS system was firstly developed by Horvath et al. in 2009, and it was structured on six categories based on 10 sonographic patterns. The authors referred to the Korean and the US TI-RADS, namely K-TIRADS and ACR-TIRADS, which are both based on 5 categories with similar but different score systems: these slight differences have consequences in nodules stratification, the associated risk of malignancies, and clinical recommendations. Please, specify the TI-RADS systems used in the study and add the correct corresponding reference.
Materials and methods
General comment: a considerable part of this section is dedicated to image processing and segmentation; however, the models proposed (Artificial Neural Network (ANN), and Support Vector Machine (SVM)), as well as the statistical analyses, are poorly described. These parts are crucial to support the manuscript results and have to be improved.
- Page 2, first paragraph, lines 42-44: It is not clear how the literature review performed impacts the models’ construction and the manuscript overall; please, explain this part properly and contextualize it within the analyses performed.
- Page 3, “2.2. Image Analysis” paragraph, lines 4: please, explain what the “further adjustments” are or, if elsewhere described, report here what/when/where these adjustments were performed.
- Page 5, “2.5. Classification” paragraph, lines 1-5: a training and a testing phase are reported as part of the procedure, but no information is provided about how the database images were subdivided into these two phases.
- Page 5, “2.5. Classification” paragraph, lines 18-19: feature extraction and training class are described, but no/little information is provided on the Artificial Neural Network and the Support Vector Machine models. Please, further described these models, possibly in two specific paragraphs.
- Page 5, “2.5. Classification” paragraph, lines 13-15: four models were built, but no description about them is provided; the reason why only two models were then chosen has to be properly and extensively described (in the Results and Discussion sections).
- Material and methods and Results, Page 8, “3.3. Graphic User Interface (GUI) from the Support Vector Machine (SVM)” paragraph, lines 1: no/little information is provided on the graphic user interface development; in figure 11, 12, and 13 there are 4 features reported, but no explanation is provided why these four features were chosen, and others were discarded (e.g., size, shape). Furthermore, if the tool developed should aid radiologist interpretations, the model should also provide a TI-RADS score in the graphic user interface based on the features reported. Finally, it is unclear why the graphic user interface is reported for the Support Vector Machine model only, and no graphic user interface for the Artificial Neural Network is presented. Was it developed for the sole SVM model? Please, further explain and discuss the graphic user interface.
Results
General comment: although the study mainly focused on the development of a Computer-aided diagnostic system, the database used for the analyses (Digital database of thyroid ultrasound images of the Universidad Nacional de Colombia) also provide data on patients age and gender, and nodules features as well; present these data in a proper paragraph and/or table of the results is strongly advised.
- Page 8, “3.3. Graphic User Interface (GUI) from the Support Vector Machine (SVM)” paragraph, lines 1: this paragraph is almost entirely composed of figures; a proper presentation of the GUI results (see the general comment of Material and methods) is recommended.
Discussion
General comment: several studies on Computer-aided diagnostic systems (in general and in the setting of thyroid nodules diagnostic) are already present in the literature, but they are almost completely neglected. A more comprehensive and critical discussion of the literature is mandatory: discuss the proposed method's outcomes and performances and compare them with the others already available.
Figures
General comment: improve figures’ quality; the resolution is inadequate (at least in the PDF version) to clearly evaluate them (e.g., in Figure 11, 12, and 13, the writings are barely legible).
References
General comment: correct the references reported (reference number 18 is listed, but it is the same as number 12) and add additional references (see Introduction and Discussion General comments).
Author Response
School of Medicine,
National University of Ireland Galway,
Galway,
Ireland
Editor,
Medicina
Date typed: May 10th, 2021,
Dear Editor,
Thank you for considering out manuscript. We found the reviewer comments helpful and constructive. Below we outline our response to each of the comments and the relevant changes made to the manuscript. if you find these satisfactory , please consider the manuscript further for publication in your journal.
Response to comments please find attached Word document.
We hope you find these responses to the reviewer comments satisfactory and look forward to hearing your response.
Yours Sincerely,
Vijay Vyas Vadhiraj,
School of Medicine,
National University of Ireland Galway

Reviewer 3 Report
I think you have to clearly explain if sensitivity and specificity are calculated with respect to the radiologist classification or to the definitive hystological feature. If the comparison was made with the radiological evaluation, these terms would have to be more extensively explained.
Could you explain how much of image elaboration is automatically made an how many time is necessary to obtain the score?
Author Response
School of Medicine,
National University of Ireland Galway,
Galway,
Ireland
Editor,
Medicina
Date typed: May 10th, 2021,
Please find attached Word document.
We hope you find these responses to the reviewer comments satisfactory and look forward to hearing your response.
Yours Sincerely,
Vijay Vyas Vadhiraj,
School of Medicine,
National University of Ireland Galway

Round 2
Reviewer 1 Report
I have no additional remarks on the revised version.
The authors have addressed my concerns.
Reviewer 2 Report
The authors have addressed my concerns and revised their manuscript accordingly.